# Cross-Domain Adaptive Multi-Scale Representation Learning for Unified Time Series Anomaly Detection

## Abstract

Time series anomaly detection has received growing attention due to its importance in a wide range of real-world applications. However, two critical challenges remain underexplored. First, most existing methods train separate models for individual domains, which severely limits their generalization ability and neglects the potential of cross-domain anomaly detection. Second, when extending to the cross-domain setting, the inconsistency of temporal granularity across datasets makes it difficult to learn unified representations. To address these issues, we propose UniAnomaly, a cross-domain anomaly detection framework equipped with a multi-scale encoder that effectively captures temporal dependencies at different granularity. Our approach enables robust and transferable representation learning across heterogeneous datasets. Extensive experiments on multiple real-world benchmarks demonstrate that UniAnomaly consistently achieves state-of-the-art performance, highlighting the effectiveness of cross-domain multi-scale modeling for time series anomaly detection. Our code is available at https://anonymous.4open.science/r/UniAnomaly-B923.

## 1 INTRODUCTION

Time series anomaly detection plays a crucial role in various applications, including industrial monitoring, financial risk management, and healthcare. Anomaly detection models not only provide early warnings but also support decision-making and risk mitigation. However, existing methods often train separate models for each domain, focusing on domain-specific temporal features. While such models may achieve strong performance within a single domain, this approach limits their generalization ability across domains and prevents the effective utilization of potential shared anomaly patterns.

The first key challenge lies in **the lack of unified temporal representations across domains**. Most existing approaches learn domain-specific features independently, without capturing cross-domain commonalities[3]. As a result, models cannot leverage patterns shared across different application scenarios. Inspired by transfer learning in vision and language processing, pretraining on a large collection of diverse datasets can significantly improve the generalization and robustness of model performance[14]. Such multi-dataset pretraining allows models to learn domain-agnostic temporal features that transfer effectively to unseen domains, bridging the gap between heterogeneous data sources.

When using datasets from different domains, **inconsistent temporal granularity** emerges as a second key challenge. Time series from different domains vary substantially in sampling frequency and temporal resolution. Short-term spikes or abrupt anomalies are often observable only at high-frequency sampling, whereas long-term trends, seasonal effects, or periodic anomalies become more apparent at lower-frequency scales. Single-scale models typically capture anomalies at a single temporal resolution while neglecting signals at other scales, limiting their ability to detect multi-scale anomalies, as shown in Figure 1. Recent work has introduced frameworks combining adaptive bottlenecks with multi-scale structures, which effectively enhance the model's ability to detect anomalies across diverse domains[34]. However, they still do not adequately account for the varying temporal granularity present across different domains.

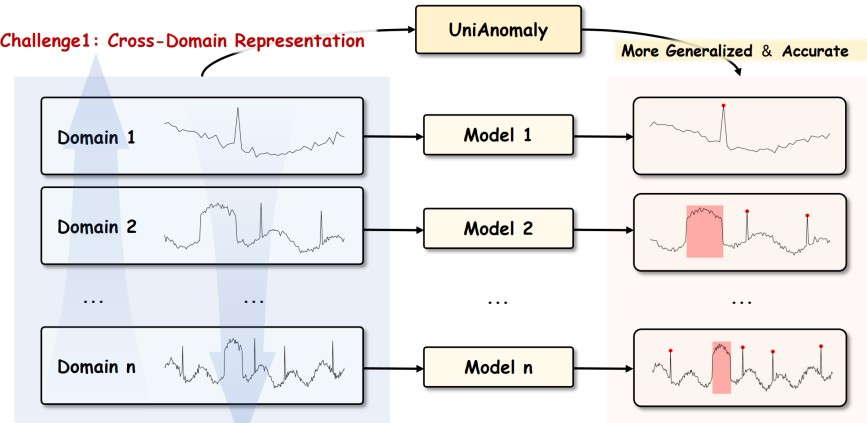

Figure 1: Challenges and Variations in Cross-Domain Anomaly Detection Methods

To address the aforementioned challenges, we propose **UniAnomaly**, a unified framework for cross-domain and multi-scale time series anomaly detection. First, to learn cross-domain temporal representations, UniAnomaly is pretrained on a large collection of datasets from diverse domains, enabling the model to capture shared anomaly patterns while retaining domain-specific information. Second, to address inconsistent temporal granularity across domains, we introduce a multi-scale architecture, where a patch embedding module encodes the input at multiple resolutions and a multi-scale encoder captures the corresponding temporal dynamics. These outputs are integrated through a scale-wise adaptive projection module, producing a unified cross-scale representation. Finally, a decoder is applied to reconstruct the original input and generate anomaly scores, enabling the model to detect anomalies across a wide range of temporal scales. Our contributions is as follows

1. To address **cross-domain representation learning**, we propose **UniAnomaly**, pretrained on diverse datasets to capture shared anomaly patterns while preserving domain-specific information.

2. To tackle **inconsistent temporal granularity across domains**, we design a **multi-scale architecture** with patch embedding, a multi-scale encoder, and a scale-wise adaptive projection module that produces unified cross-scale representations.

3. By integrating these designs, **UniAnomaly** achieves **robust anomaly detection across domains and temporal scales**, effectively handling heterogeneous datasets with varying temporal patterns.

## 2 RELATED WORK

**Time Series Reconstruction Methods.** Reconstruction-based approaches posit that models trained only on normal patterns will poorly reconstruct anomalies. Early designs used recurrent and convolutional autoencoders ([28],[46]), and variational extensions such as OmniAnomaly ([38]) introduced probabilistic reconstruction. More recent research has shifted toward masking- and diffusion-based generative frameworks, which significantly improve robustness to noise, missing values, and distribution shifts. For example, masked autoencoder variants for time series ([22],[8]) exploit temporal–frequency masking to learn context-aware representations that are harder to overfit to anomalies. Diffusion-based approaches ([43],[41]) leverage denoising and distribution augmentation to generate sharper reconstructions and more discriminative residuals. Very recently, physics-informed diffusion models ([37]) have been proposed to incorporate structural priors into reconstruction, enabling unsupervised anomaly detection in complex dynamical systems.

**Time Series Anomaly Detection Methods.** Time series anomaly detection (TSAD) has advanced significantly in recent years, evolving beyond classical statistical models and shallow machine learn-

ing methods toward deep representation learning. Transformer-based architectures have become a dominant paradigm, leveraging self-attention to capture long-range temporal dependencies and enabling both anomaly detection and localization.[42],[39] Self-supervised and contrastive learning approaches further address the scarcity of labeled anomalies by constructing proxy tasks or synthetic perturbations to learn robust normality representations[5],[47]. In parallel, graph neural networks and spatio-temporal modeling explicitly capture dependencies among multivariate signals, which is especially critical in industrial and sensor data.[6], Generative models such as VAEs[19], GANs[10], and more recently diffusion models[45] have also been employed to estimate data likelihoods or reconstruction errors, providing probabilistic criteria for anomaly scoring.

## 3 METHODOLOGY

Given a test time series $\mathcal{D}_{\text{test}} = (\mathbf{x}_1, \mathbf{x}_2, \cdots, \mathbf{x}_T) \in \mathbb{R}^{T \times C}$ with $T$ time steps and $C$ variables, the anomaly detection task aims to predict $\hat{\mathbf{y}}_{\text{test}} = (y_1, y_2, \cdots, y_T)$, where $y_t \in \{0, 1\}$ indicates whether the observation $\mathbf{x}_t \in \mathbb{R}^C$ is anomalous. We consider building a general model pre-trained on $M$ multi-domain datasets $\mathcal{D} = \{\mathcal{D}^{(i)}\}_{i=1}^M$, where each dataset $\mathcal{D}^{(i)} = (\mathbf{x}_1^{(i)}, \mathbf{x}_2^{(i)}, \cdots, \mathbf{x}_{T^{(i)}}^{(i)}) \in \mathbb{R}^{T^{(i)} \times C^{(i)}}$ contains $T^{(i)}$ observations with $C^{(i)}$ variables. The goal is to detect anomalies on unseen datasets $\mathcal{D}_{\text{test}} \notin \mathcal{D}$.

### 3.1 OVERALL ARCHITECTURE

Our framework is designed to extract cross-domain multi-scale representations for anomaly detection, as shown in Figure2. To address the issue of varying temporal granularity, we design a Multi-Scale Patch Embedding Module and a Multi-Scale Encoder to extract features at different scales, which segments the input into patches of multiple lengths and maps them into a latent space, enabling simultaneous modeling of short-term and long-range dependencies. To obtain a unified representation that adapts across domains, we introduce the Scale-Wise Adaptive Projection Module, which adaptively fuses multi-scale features by reweighting their contributions in a context-aware manner. Finally, to effectively learn normal patterns, we employ a decoder module consisting of a multi-layer perceptron (MLP), and use the variance of the reconstructed values as the anomaly score.

### 3.2 MULTI-SCALE PATCH EMBEDDING

To capture temporal patterns at multiple granularity, we propose a **Multi-Scale Patch Embedding (MSPE)** module, which extracts hierarchical features from input sequences through multi-scale patching and embedding.

**Multi-Scale Patch Generation.** We first define a set of patch sizes $L_1, L_2, \ldots, L_S$ corresponding to different temporal resolutions. The input sequence $\mathbf{x} \in \mathbb{R}^{T \times D}$ is then segmented into non-overlapping patches for each scale, producing a collection of multi-scale patches:

$$\mathbf{p}_s = \text{Patch}_{L_s}(\mathbf{x}), \quad s = 1, \ldots, S, \tag{1}$$

where $\mathbf{p}_s$ contains patches of length $L_s$ and captures patterns at the corresponding temporal scale. This design allows the model to simultaneously represent both fine-grained fluctuations and long-term dependencies in the sequence.

**Multi-Scale Patch Embedding.** Each patch at scale $L_s$ is first projected into a latent embedding space through a linear transformation, after which a random masking operation is applied to partially drop patch tokens. This masking strategy encourages the model to learn more robust temporal representations by preventing over-reliance on specific patches and enhancing generalization across different scales.

$$\mathbf{h}_s = \text{Linear}(\mathbf{p}_s), \quad s = 1, \ldots, S. \tag{2}$$

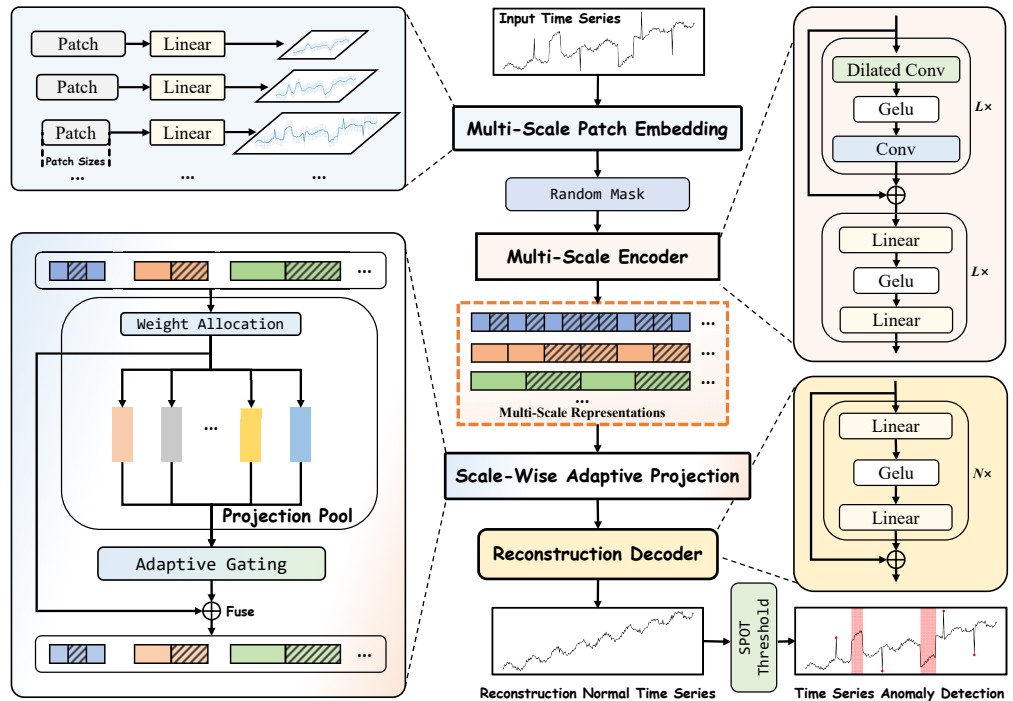

Figure 2: Model Architecture. UniAnomaly comprises four main modules: the Multi-Scale Patch Embedding Module(**MSPE**) and the Multi-Scale Encoder Module(**MSEM**) are employed to capture features at different scales. The Scale-Wise Adaptive Projection Module(**SAPM**) is designed to integrates these multi-scale features to obtain a generalized representation. Finally, the Reconstruction Decoder is used to reconstruct the input sequences and compute anomaly scores. The final time series anomaly detection results are obtained by integrating the outputs with Spot algorithm.

The resulting $\mathbf{h}_s$ forms a multi-scale representation of the input sequence, where each scale encodes distinct hierarchical features of the temporal structure. By incorporating information from these diverse scales, the model effectively leverages complementary cues across multiple temporal resolutions.

## 3.3 MULTI-SCALE ENCODER

To achieve comprehensive characterization of features at multiple scales and enable effective cross-dataset temporal modeling, we designed a Multi-Scale Encoder module. This module processes the multi-scale embeddings to extract hierarchical multi-scale representations.

To effectively capture temporal patterns across different resolutions, the multi-scale embeddings $\mathbf{h}_s$ generated by **MSPE** are independently processed by scale-specific encoders. Each encoder is instantiated as a dilated convolutional network with residual connections, a design that facilitates modeling both short-term dynamics and long-range dependencies while maintaining computational efficiency. Concretely, the encoder comprises $L$ layers of dilated convolutions interleaved with GELU activations and layer normalization, formally defined for layer $l = 1, \ldots, L$:

$$\mathbf{h}_s^{(l)} = \text{LayerNorm}\Big(\mathbf{h}_s^{(l-1)} + \text{GELU}\big(\text{Conv}(\mathbf{h}_s^{(l-1)})\big)\Big), \tag{3}$$

where $\mathbf{h}_s^{(0)}$ is the input embedding at scale $s$, and $\mathbf{h}_s^{(l)} \in \mathbb{R}^{P \times R}$ is the hidden representation after the $l$-th convolutional layer, with $R$ being the hidden dimension of the encoder. Here, $\text{Conv}(\cdot)$ denotes a dilated convolution, $\text{GELU}(\cdot)$ is the Gaussian Error Linear Unit activation function introducing nonlinearity, and $\text{LayerNorm}(\cdot)$ normalizes the features along the channel dimension to stabilize training. After $L$ layers, the encoder produces the final output $\mathbf{h}_s^{(L)} \in \mathbb{R}^{P \times R}$, which serves as the

**scale-specific hierarchical feature**. This representation preserves the multi-resolution structure of the input sequence while effectively capturing temporal patterns unique to scale $s$, thereby enabling the encoder to model both fine-grained fluctuations and long-range dependencies.

## 3.4 SCALE-WISE ADAPTIVE PROJECTION

The multi-scale features produced by the **MSEM** capture temporal dynamics at different resolutions, but their information content and relevance vary across scales, making naive aggregation suboptimal. To address this, we introduce a Scale-Wise Adaptive Projection Module **(SAPM)**, which consists of two components: (i) an Adaptive Projection Pool that refines each scale-specific feature into a more comparable representation space, and (ii) an Adaptive Gating Mechanism that dynamically assigns weights to different scales for fusion. By aggregating the scales in this adaptive manner, the module produces multi-scale features with improved generalization, emphasizing informative scales while suppressing redundant ones.

**Adaptive Projection Pool.**    To handle variations across temporal scales and enable robust cross-domain generalization, we introduce a Scale-Wise **A**daptive **P**rojection **P**ool (APP). Different temporal scales often vary in both information content and noise characteristics: some scales carry essential patterns, while others may introduce redundant or noisy signals. Simply aggregating all scales without adaptation risks overfitting to irrelevant details or suppressing critical cues.

Denote the output of the Multi-Scale Encoder as $\{\mathbf{h}_s\}_{s \in \mathcal{S}}$, where $\mathbf{h}_s \in \mathbb{R}^{P_s \times R}$ represents the hierarchical features at scale $s$, with $P_s$ patches and feature dimension $R$. The collection of multi-scale features can thus be written as $\mathbf{H} = \{\mathbf{h}_s\}_{s=1}^{S}$. Each scale-specific projection pool is defined as:

$$\mathrm{APP}_s(\mathbf{h}_s) = \mathbf{h}_s + \mathrm{MLP}_s^{(N)}(\mathbf{h}_s), \tag{4}$$

where $\mathrm{MLP}_s^{(N)}$ denotes a residual multilayer perceptron with $N$ layers, each consisting of linear transformations, GELU activations, and dropout. By adapting the latent representation individually for each scale, the module effectively retains salient temporal patterns while suppressing irrelevant noise.

**Adaptive Gating Mechanism.**    Building on the outputs of the **APP**, we further introduce a learnable gating mechanism to adaptively weight the contribution of each temporal scale. Even after scale-specific adaptation, the relative importance of different scales may vary depending on the input sequence, making it crucial to emphasize informative scales while down-weighting less relevant ones.

For each scale $s$, the gating network computes a relevance score:

$$\mathbf{g}_s = \mathrm{Gate}(\mathrm{APP}_s(\mathbf{h}_s)) \in \mathbb{R}^{P_s \times 1}, \tag{5}$$

where $\mathrm{Gate}(\cdot)$ is a compact MLP mapping the feature dimension $R$ of $\mathbf{h}_s$ to a scalar score for each patch. The scores from all scales are concatenated and normalized via softmax, and the final fused representation is obtained as:

$$\mathbf{W} = \mathrm{softmax}(\mathrm{cat}(\{\mathbf{g}_s\}_{s \in \mathcal{S}})) \in \mathbb{R}^{P_{\max} \times S}, \tag{6}$$

$$\mathbf{h}_{\mathrm{fused}} = \sum_{s \in \mathcal{S}} \mathbf{W}_{:,s} \odot \mathrm{APP}_s(\mathbf{h}_s), \tag{7}$$

where $\mathcal{S}$ is the number of scales, $P_{\max}$ is the maximum number of patches across scales, and $\odot$ denotes element-wise multiplication with broadcasting. Here, $\mathrm{cat}(\cdot)$ concatenates the relevance scores from all scales along the scale dimension, and softmax normalizes them.

Finally, the fused multi-scale representation $\mathbf{h}_{\mathrm{fused}}$ is fed into a reconstruction module, which is implemented as an $N$-layer MLP, to reconstruct the input sequence. The reconstruction error, measured as the variance between the reconstructed and original sequences, serves as the **anomaly score** for

| Dataset | SMD | | | MSL | | | SMAP | | | SWaT | | | PSM | | |
|---|---|---|---|---|---|---|---|---|---|---|---|---|---|---|---|
| Metric | P | R | F1 | P | R | F1 | P | R | F1 | P | R | F1 | P | R | F1 |
| OCSVM | 66.98 | 82.03 | 73.75 | 50.26 | 99.86 | 66.87 | 41.05 | 69.37 | 51.58 | 56.80 | 98.72 | 72.11 | 57.51 | 58.11 | 57.81 |
| PCA | 64.92 | 86.06 | 74.01 | 52.69 | 98.33 | 68.61 | 50.62 | 98.48 | 66.87 | 62.32 | 82.96 | 71.18 | 77.44 | 63.68 | 69.89 |
| HBOS | 60.34 | 64.11 | 62.17 | 59.25 | 83.32 | 69.25 | 41.54 | 66.17 | 51.04 | 54.49 | 91.35 | 68.26 | 78.45 | 29.82 | 43.21 |
| LOF | 57.69 | 99.10 | 72.92 | 49.89 | 72.18 | 59.00 | 47.92 | 82.86 | 60.72 | 53.20 | 96.73 | 68.65 | 53.90 | 99.91 | 70.02 |
| IForest | 71.94 | 94.27 | 81.61 | 53.87 | 94.58 | 68.65 | 41.12 | 68.91 | 51.51 | 53.03 | 99.95 | 69.30 | 69.66 | 88.79 | 78.07 |
| LODA | 66.09 | 84.37 | 74.12 | 57.79 | 95.65 | 72.05 | 51.51 | 100.00 | 68.00 | 56.30 | 70.34 | 62.54 | 62.22 | 87.38 | 72.69 |
| AE | 69.22 | 98.48 | 81.30 | 55.75 | 96.66 | 70.72 | 39.42 | 70.31 | 50.52 | 54.92 | 98.20 | 70.45 | 60.67 | 98.24 | 75.01 |
| DAGMM | 63.57 | 70.83 | 67.00 | 54.07 | 92.11 | 68.14 | 50.75 | 96.38 | 66.49 | 59.42 | 92.36 | 72.32 | 68.22 | 70.50 | 69.34 |
| LSTM | 60.12 | 84.77 | 70.35 | 58.82 | 14.68 | 23.49 | 55.25 | 27.70 | 36.90 | 49.99 | 82.11 | 62.15 | 57.06 | 95.92 | 71.55 |
| BeatGAN | 74.11 | 81.64 | 77.69 | 55.74 | 98.94 | 71.30 | 54.04 | 98.30 | 69.71 | 61.89 | 83.46 | 71.08 | 58.81 | 99.08 | 73.81 |
| Omni | 79.09 | 75.77 | 77.40 | 51.23 | 99.40 | 67.61 | 52.74 | 98.51 | 68.70 | 62.76 | 82.82 | 71.41 | 69.20 | 80.79 | 74.55 |
| CAE-Ensemble | 73.05 | 83.61 | 77.97 | 54.99 | 93.93 | 69.37 | 62.32 | 64.72 | 63.50 | 62.10 | 82.90 | 71.01 | 73.17 | 73.66 | 73.42 |
| MEMTO | 49.69 | 98.05 | 65.96 | 52.73 | 97.34 | 68.40 | 50.12 | 99.10 | 66.57 | 56.47 | 98.02 | 71.66 | 52.69 | 83.94 | 64.74 |
| A.T. | 54.08 | 97.07 | 69.46 | 51.04 | 95.36 | 66.49 | 56.91 | 96.69 | 71.65 | 53.63 | 98.27 | 69.39 | 54.26 | 82.18 | 65.37 |
| DCdetector | 50.93 | 95.57 | 66.45 | 55.94 | 95.53 | 70.56 | 53.12 | 98.37 | 68.99 | 53.25 | 98.12 | 69.03 | 54.72 | 86.36 | 66.99 |
| SensitiveHUE | 60.34 | 90.13 | 72.29 | 55.92 | 98.95 | 71.46 | 53.63 | 98.37 | 69.42 | 58.91 | 91.71 | 71.74 | 56.15 | 98.75 | 71.59 |
| D3R | 64.87 | 97.93 | 78.04 | 66.85 | 90.83 | 77.02 | 61.76 | 92.55 | 74.09 | 60.14 | 97.57 | 74.41 | 73.32 | 88.71 | 80.29 |
| ModernTCN | 74.07 | 94.79 | 83.16 | 65.94 | 93.00 | 77.17 | 69.50 | 65.45 | 67.41 | 59.14 | 89.22 | 71.13 | 73.47 | 86.83 | 79.59 |
| GPT4TS | 73.33 | 95.97 | 83.14 | 64.86 | 95.43 | 77.23 | 63.52 | 90.56 | 74.67 | 56.84 | 91.46 | 70.11 | 73.61 | 99.08 | 81.44 |
| DADA(zero shot) | 76.50 | 94.54 | 84.57 | 68.70 | 91.51 | 78.48 | 65.85 | 88.25 | 75.42 | 61.59 | 94.59 | 74.60 | 74.31 | 92.11 | 82.26 |
| **UniAnomaly**(zero shot) | 75.21 | 97.38 | **84.87** | 69.50 | 92.85 | **79.50** | 65.28 | 89.70 | **75.56** | 63.95 | 91.80 | **75.38** | 77.34 | 93.63 | **84.68** |

Table 1: Results for five real-world datasets.

each time point. To detect anomalies, we follow the SPOT method (**?**) to compute a threshold $\delta$, and a time point is flagged as anomalous if its anomaly score exceeds $\delta$. This procedure effectively leverages the multi-scale, generalized features learned by the encoder and adaptive projection modules, highlighting temporal regions that deviate from normal patterns.

## 4 EXPERIMENTS

### 4.1 EXPERIMENTAL SETTINGS

**Datasets.** In the pre-training stage, we use a cross-domain corpus for time series reconstruction by combining widely used time series datasets, including ASD[20] , Exathlon[16], ECG[27] and MITDB[27], OPP[31], SVDB[12], GAIA[26], IOPS[30], MGAB[23], NYC[4], SKAB[17], YA-HOO[13], together with diverse time series from the Monash Prediction Library[11]. To demonstrate the effectiveness of our method, we evaluate widely used benchmark datasets including SMD[38], MSL[15], SMAP[15], SWaT[25] and PSM[1].

**Baselines.** We compare our model with 20 baselines for comprehensive evaluations, including the linear transformation-based models: OCSVM[33], PCA[35], the density estimation-based methods: HBOS[7], LOF[2], the outlier-based methods: IForest[21], LODA[29]; the neu- ´ral network-based models: AutoEncoder[32], DAGMM[50], LSTM[15], CAE-Ensemble[32], BeatGAN[48], Omni-Anomaly (Omni)[38], Anomaly Transformer (A.T.)[42], MEMTO[36], DCdetector[44], D3R[40], GPT4TS[49], ModernTCN[24], SensitiveHUE[9], DADA[34].

**Metrics.** Following recent studies[34], We adopt Precision (P), Recall (R), F1-score (F1), and AUC-ROC (AUC) as evaluation metrics for time series anomaly detection. For the F1-score, we use the affiliation-based variant rather than the commonly applied Point Adjustment, since the latter tends to overestimate performance by labeling an entire anomaly segment as detected when only a single point is correctly identified.

### 4.2 MAIN RESULTS

**Zero-Shot Results.** We evaluate UniAnomaly against a variety of baselines across multiple anomaly detection datasets, as summarized in Table 4.2. Different from many non-cross-domain approaches that rely on dataset-specific training and often fail to generalize beyond their source domain, UniAnomaly follows a cross-domain pretraining protocol and directly transfers to diverse downstream datasets, demonstrating robust generalization without sacrificing performance. Across all benchmarks, UniAnomaly consistently achieves the best F1 scores, surpassing methods that train separately on each dataset. This highlights its ability to learn domain-invariant temporal representations that remain effective even in unseen settings. Moreover, a key advantage of UniAnomaly lies in its multi-scale feature extraction: while single-scale models struggle to capture both short-term

| Dataset | Method | F1 | AUC-ROC |
|---|---|---|---|
| SMD | A.T | 66.42 | 50.02 |
| | D3R | 78.02 | 53.34 |
| | GPT4TS | 83.13 | 71.15 |
| | ModernTCN | 83.16 | 70.21 |
| | **UniAnomaly(finetune)** | **85.01** | **72.11** |
| MSL | A.T | 66.49 | 34.05 |
| | D3R | 77.02 | 45.77 |
| | GPT4TS | 77.23 | 72.63 |
| | ModernTCN | 77.17 | 74.96 |
| | **UniAnomaly(finetune)** | **79.82** | **76.33** |
| PSM | A.T | 65.37 | 50.11 |
| | D3R | 80.29 | 50.22 |
| | GPT4TS | 81.44 | 58.94 |
| | ModernTCN | 79.59 | 58.46 |
| | **UniAnomaly(finetune)** | **85.61** | **63.90** |

Table 2: Comparison of different methods on three benchmark datasets. The best results are highlighted in bold.

| Method | PSM | | SMAP | | SWAT | |
|---|---|---|---|---|---|---|
| | F1 | AUC | F1 | AUC | F1 | AUC |
| UniAnomaly (Routing) | 84.02 | 63.21 | 70.78 | 50.86 | **75.31** | **81.79** |
| UniAnomaly (Gating) | **84.67** | **63.43** | **71.77** | **51.42** | 75.29 | 81.49 |

Table 3: Comparison performance of MoE and Gate based UniAnomaly on three benchmark datasets.

fluctuations and long-term dependencies, our design explicitly models multiple temporal resolutions. Even when pretraining data overlap with that of strong baselines such as DADA, UniAnomaly achieves an additional 1–2% improvement in F1, showing that multi-scale modeling is indispensable for robust anomaly detection. Together, these results demonstrate that UniAnomaly not only generalizes effectively across domains but also benefits from a principled multi-scale design that yields state-of-the-art performance across diverse anomaly detection scenarios.

**Finetune Results.** We further evaluate the capability of UniAnomaly after finetuning, with results reported on the SMD, MSL, and PSM benchmarks in Table 4.2. UniAnomaly achieves the best performance across all three datasets, underscoring its strong ability to generalize when adapted to specific domains. Beyond absolute score improvements, the finetuned model consistently outperforms prior approaches in both F1 and AUC-ROC, demonstrating that the cross-domain representations learned during pretraining can be effectively specialized without overfitting. These improvements can be attributed to the proposed multi-scale representation and adaptive fusion, which together allow the model to disentangle informative temporal patterns from noise.

## 4.3 MODEL ANALYSIS

In this section, we systematically evaluate the effectiveness of UniAnomaly's multi-scale design, including the Multi-scale Patch Embedding and Multi-scale Encoder modules, as well as the gating mechanisms within Adaptive Projection module. We also conduct ablation studies to examine the impact of key hyperparameters such as input sequence length (window size), unified representation feature dimension, and the number of temporal scales on anomaly detection performance, providing a comprehensive understanding of how each component contributes to the model's effectiveness and robustness.

**Analysis on gating mechanism.** In UniAnomaly, we incorporate a Gating Mechanism within the Adaptive Projection module to enhance feature selection. To assess its effectiveness, we compare it with the Routing Mechanism employed in MoE[18].

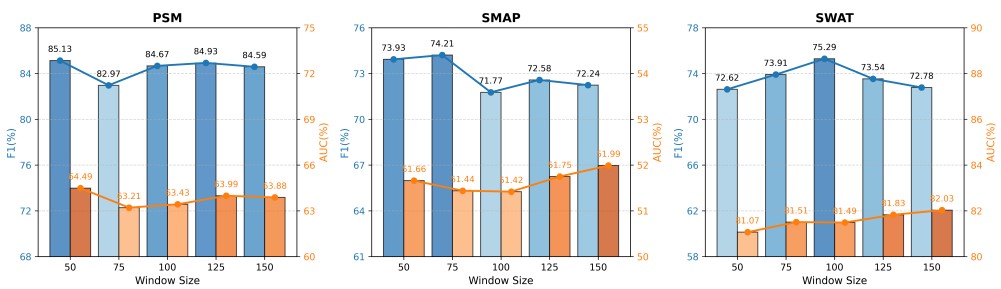

Figure 3: Model performance across datasets with different window sizes.

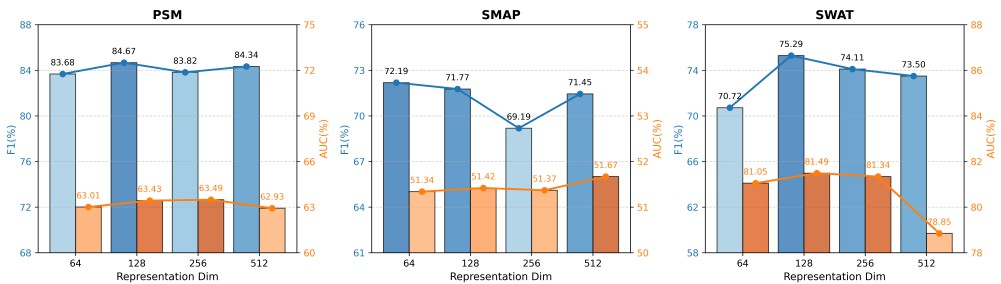

Figure 4: Model performance across datasets with different representation dims.

As shown in Table 4.3, the gating-based UniAnomaly yields consistent improvements over the routing-based variant on PSM and SMAP, with F1 improvements of 0.65% (84.02% → 84.67%) and 0.99% (70.78% → 71.77%) and corresponding AUC gains of 0.22% (63.21% → 63.43%) and 0.56% (50.86% → 51.42%), respectively. This demonstrates that the gating mechanism is more effective at aggregating multi-scale features for anomaly detection on these datasets. On SWAT, both modules exhibit similar performance, indicating that the benefits of gating are less pronounced on datasets with simpler temporal patterns. Overall, these results underscore the contribution of gating-based feature fusion in enhancing UniAnomaly's generalization across diverse anomaly detection domains.

**Analysis on window size.** We further investigate the impact of input sequence length (window size) on model performance across three benchmark datasets, as shown in Figure 3. Although Uni-Anomaly adopts a multi-scale architectural design, datasets from different domains exhibit varying sensitivities to the choice of window size. Specifically, on the PSM dataset, both F1 and AUC reach their peaks at a relatively short window size of 50, suggesting that PSM benefits more from local temporal representations, where short-term dependencies are sufficient to capture most anomaly patterns. In contrast, for SMAP and SWAT, the optimal F1 scores are achieved at window sizes of 75 and 100, respectively, while AUC continues to increase and reaches its maximum at 150. This divergence indicates that longer windows tend to provide more stable global representations, which are favored by AUC since it evaluates ranking consistency across thresholds, whereas F1 is more sensitive to precise decision boundaries and thus degrades with excessively long windows. Overall, these results highlight the domain-dependent inconsistency of temporal granularity and further demonstrate the advantage of the gating mechanism in adaptively selecting features across different scales.

**Analysis on representation dim.** We examine the effect of representation dimension on model performance, as shown in Figure 4. On PSM, both F1 and AUC remain relatively stable across different dimensions, with the best F1 achieved at 128. In contrast, SMAP exhibits little improvement as the dimension grows, and its performance even slightly degrades, suggesting that this dataset is less sensitive to representation dimensionality. For SWAT, increasing the dimension leads to a clear gain in F1, with the peak observed at 128, but further enlargement to 512 results in a drop in AUC, indicating that overly large dimensions may introduce redundant features and hurt generalization.

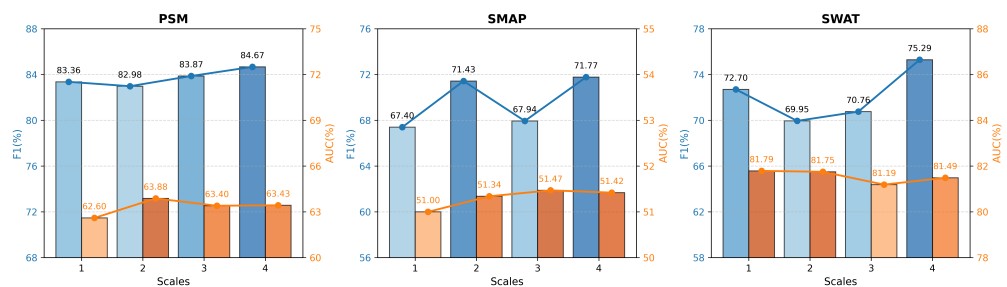

Figure 5: Model performance across datasets with different scales.

Overall, these results highlight that the optimal representation dimension is dataset-dependent: PSM and SWAT benefit from moderate dimensions, while SMAP remains relatively robust. This suggests that careful selection of representation dimensionality is crucial for achieving the best trade-off between expressive capacity and generalization.

We examine the effect of representation dimensionality on model performance, as shown in Figure 4. For PSM and SMAP, changes in dimension have only minor effects, with F1 and AUC varying within 1–3%. On PSM, the best F1 is observed at 128, while on SMAP the highest F1 occurs at 64, and further increases in dimension do not lead to consistent improvements. In contrast, SWAT is more sensitive, with performance differences reaching 3–5%, and both F1 and AUC peaking at 128. Notably, when the dimension is enlarged to 512, AUC drops substantially, suggesting that overly large representations may introduce redundancy and hurt generalization. This trend aligns with Figure 3, where the best F1 on SWAT appears at a window size of 100 rather than 150, indicating that excessive representational capacity does not necessarily translate into better detection performance. Overall, these results indicate that the optimal dimensionality varies across datasets. PSM and SMAP are relatively robust to changes in dimension, whereas SWAT shows clear improvements with moderate dimensions such as 128. Moreover, the differing sensitivities reflect domain-specific variations in information density and temporal granularity, which underscores the complexity of cross-domain anomaly detection. These observations provide further motivation for adopting a unified multi-scale modeling strategy.

**Analysis on scales.** We also investigate the impact of scales, which control the granularity of time-series patches used for representation (see Appendix for a detailed description). As shown in Figure 5, the overall AUC remains relatively stable across scales, whereas F1 is more sensitive and reaches its best at scale 4. Importantly, performance does not always improve monotonically with richer scales. For PSM, both F1 and AUC are stable with only minor variations (about 1–2%), while SMAP and SWAT exhibit larger fluctuations of up to 3–5%, indicating stronger scale effects and higher sensitivity in these domains. In particular, SMAP shows a drop in F1 when moving from scale 2 to 3, and SWAT decreases from scale 1 to 2, suggesting that smaller scales may capture only locally optimal patterns for a given dataset. Nevertheless, as the scale further increases, F1 improves again, with the best results obtained at scale 4. These findings demonstrate that multi-scale representations are essential for cross-domain anomaly detection, since different domains favor different temporal granularity.

## 5 CONCLUSION

This paper presents UniAnomaly, a unified framework for time series anomaly detection that directly addresses two long-standing challenges in the field: limited cross-domain generalization and inconsistent temporal granularity. To this end, we design a multi-scale encoder that captures temporal dependencies at different levels of granularity, enabling robust and transferable representation learning across heterogeneous datasets. Extensive experiments on multiple real-world benchmarks demonstrate that UniAnomaly consistently achieves state-of-the-art performance, highlighting cross-domain multi-scale modeling as an effective solution for advancing time series anomaly detection.

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

iclr2026$_c$on$ference$

# A    APPENDIX

## A.1    EXPERIMENT SETTING DETAILS

### A.1.1    DATASET DESCRIPTIONS

In this work, we utilize both pre-training datasets and validation datasets. The pre-training datasets are employed to provide diverse temporal and domain knowledge, while the validation datasets are used to assess the generalization capability and robustness of our proposed method. A summary of the datasets is presented in Tables 5 and 4, respectively.

| Dataset | Domain | Dimension (C) | Anomaly Ratio (AR) |
|---------|--------|---------------|--------------------|
| SMD | Server Machine | 38 | 4.2% |
| MSL | Spacecraft | 1 | 10.5% |
| SMAP | Spacecraft | 1 | 12.8% |
| SWaT | Water treatment | 31 | 12.1% |
| PSM | Server Machine | 25 | 27.8% |

Table 4: Summary of evaluation datasets with their domains, dimensions, and anomaly ratios (AR).

### A.1.2    EVALUATION METRICS

We report four widely used metrics for time series anomaly detection: **Precision (P)**, **Recall (R)**, **F1-score (F1)**, and **AUC-ROC (AUC)**. Given true positives (TP), false positives (FP), and false negatives (FN), the metrics are defined as:

$$P = \frac{TP}{TP + FP}, \quad R = \frac{TP}{TP + FN}, \quad F1 = \frac{2 \times P \times R}{P + R}. \tag{8}$$

| Dataset | Domain | Dimension (C) | AR (%) |
|---------|--------|---------------|--------|
| ASD | Application Server | 19 | 1.55 |
| Exathlon | Application Server | multi | 8.71 |
| ECG | Health | 1 | 4.70 |
| MITDB | Health | 1 | 3.44 |
| OPP | Health | 1 | 4.11 |
| SVDB | Health | 1 | 4.68 |
| GAIA | AIOps | 1 | 1.21 |
| IOPS | Web | 1 | 2.15 |
| MGAB | Mackey-Glass | 1 | 0.20 |
| NYC | Transport | 3 | 0.57 |
| SKAB | Machinery | 8 | 3.65 |
| YAHOO | Multiple | 1 | 0.62 |
| Monash | Multiple | 1 | - |

Table 5: Summary of pretraining datasets with their domains, dimensions, and anomaly ratios (AR).

The F1-score is computed using the **affiliation-based variant** rather than the commonly adopted Point Adjustment. While Point Adjustment counts an entire anomaly segment as correctly detected if only a single point is captured, the affiliation-based approach evaluates anomaly segments based on their actual overlap with predictions, providing a stricter and more faithful assessment of detection quality.

For AUC-ROC, we compute the area under the Receiver Operating Characteristic curve, which plots the True Positive Rate (TPR) against the False Positive Rate (FPR):

$$TPR = \frac{TP}{TP + FN}, \quad FPR = \frac{FP}{FP + TN}. \tag{9}$$

A.2 IMPLEMENTATION DETAILS

**Model details.** The input sequence length is fixed to a window size of 100. For the multi-scale patch embedding, we use patch lengths $\{2, 4, 8, 16\}$, and the hidden dimensions are set to $\{64, 96, 128, 256\}$, with the unified representation dimension fixed at 128. Besides, the correspondence between scales and dimensions is summarized in Table 6. The Adaptive Projection Pool dimensions are chosen from $\{16, 32, 64, 128, 192, 256\}$, and the encoder depth is 5 layers. We adopt a random masking strategy to enhance robustness.

| Scales | Patch length | Hidden dim |
|--------|--------------|------------|
| 1 | [16] | [256] |
| 2 | [8, 16] | [128, 256] |
| 3 | [4, 8, 16] | [96, 128, 256] |
| 4 | [2, 4, 8, 16] | [64, 96, 128, 256] |

Table 6: Correspondence between the number of scales, patch lengths, and hidden dimensions.

**Training details.** We summarize all hyper-parameters as follows. We pretrain the model for 10 epochs with a batch size of 4096, using a 9:1 split between training and validation. The AdamW optimizer is adopted with a linearly decaying learning rate schedule, starting from $1 \times 10^{-3}$ and gradually decreasing to $4 \times 10^{-4}$, with a warm-up period of 200 iterations. We apply a sliding window of size 100 and conduct anomaly detection using non-overlapping windows, consistent with prior works. All experiments are conducted using PyTorch on a single NVIDIA A100-80GB GPU.

**Comparisons between UniAnomaly and existing methods.** As shown in Table 7, existing anomaly detection methods exhibit different limitations. Common AD methods support multi-scale modeling but fail to generalize in zero-shot scenarios or across domains. DADA achieves zero-shot and cross-domain capabilities but does not handle multi-scale temporal variations. In contrast, Uni-Anomaly unifies all three aspects, enabling zero-shot application, effective cross-domain transfer,

and multi-scale feature modeling. This combination highlights UniAnomaly's superior versatility and practical applicability across diverse time series datasets.

| Method | Zero-shot | Cross-domain | Multi-scale |
|---|---|---|---|
| Common AD Methods | × | × | ✓ |
| DADA | ✓ | ✓ | × |
| UniAnomaly | ✓ | ✓ | ✓ |

Table 7: Comparison of UniAnomaly and existing anomaly detection methods.

## A.3 LARGE LANGUAGE MODEL (LLM) USAGE STATEMENT

In preparing this paper, we utilized large language models (LLMs) to assist with aspects of manuscript writing, including language refinement, grammar corrections, and partial code implementation. The LLMs provided support in drafting and polishing text and code snippets; however, all scientific ideas, experimental design, results, and conclusions reported in this paper are solely the responsibility of the authors. We have ensured that all contributions from LLMs are acknowledged, and the final content, accuracy, and integrity of the paper remain fully under the authors' control.

