# OpenReview forum: "Cross-Domain Adaptive Multi-Scale Representation Learning for Unified Time Series Anomaly Detection"
_ICLR.cc/2026/Conference — Submitted to ICLR 2026_

### Official Review · Reviewer_ngVn · 2025-10-24

**Soundness:** 2
**Presentation:** 2
**Contribution:** 2
**Rating:** 2
**Confidence:** 4

**Summary:**

This work proposes UniAnomaly, a novel approach enabling cross-domain time series anomaly detection through large-scale pre-training. Additionally, the authors introduce a multi-scale encoder to capture patterns across different granularities. Extensive experiments demonstrate that the method achieves state-of-the-art (SOTA) performance on multiple benchmark datasets.

**Strengths:**

The authors attempt to enhance the generalization ability of anomaly detection methods through pre-training, which holds significant practical implications. Additionally, they integrate multi-scale techniques into the training process, which is equally crucial for time series anomaly detection tasks.

**Weaknesses:**

1. Insufficient innovation. Recent works [1][2] have performed multi-scale modeling from the perspective of dividing patches of different sizes. What are the differences between yourmethod and theirs?

[1] Cho Y, Lee J Y. CoMRes: Semi-Supervised Time Series Forecasting Utilizing Consensus Promotion of Multi-Resolution. In ICLR, 2025.

[2] Woo G, Liu C, Kumar A, et al. Unified training of universal time series forecasting transformers. In ICML, 2024.

2. The experiments are incomplete. No ablation studies have been conducted to verify the effectiveness of the proposed method.

3. Why is dilated convolution adopted as the encoder instead of transformer? As a pre-trained model, does it offer better scalability than transformer? Relevant experiments or analyses would help clarify this question.

4. The “Reconstruction error” in line 268 lacks a crucial explanation.

5. I recommend that the authors refine the formatting of the paper, including but not limited to the following aspects:

    a) In line 110 and 114, the citations should be placed inside the period.

    b) There is a citation issue in line 286.

    c) Line 317 should cite Table 1.

    d) The variable P mentioned in Line 211 lacks an explanation.

    e) Equations (4) and (5) lack explanations of the MLP and Gate components. What is the role of the MLP here? Does it involve dimension transformation? Providing dimension information would facilitate a better understanding of this section. The same applies to the Gate.

**Questions:**

1. What is the meaning of “shared anomaly patterns” in Line 75? Is the model learning anomaly patterns or normal patterns? Does the method use anomalous data for training?

2. Is the F1-score in Table 2 the affiliation F1-score? Why do the result, like A.T on SMD,  differ from those in Table 1?

3. Compared with DADA, UniAnomaly captures multi-scale information. Why is multi-scale information necessary for pre-trained models?

---

### Official Review · Reviewer_7Tgc · 2025-10-31

**Soundness:** 2
**Presentation:** 2
**Contribution:** 2
**Rating:** 2
**Confidence:** 4

**Summary:**

This work presents a network primarily based on reconstruction error for cross-domain time series anomaly detection learning. The authors emphasize that the challenge in cross-domain temporal anomaly detection tasks lies in inconsistent temporal granularity. To address this, they propose a gating mechanism to achieve information integration across multiple granular patch representations.

**Strengths:**

While cross-domain anomaly detection is somewhat interesting, I do not see the advantages of the proposed method or a reasonable mechanism for cross-domain adaptation.

**Weaknesses:**

Regarding the motivation, I find it unclear why multi-scale reconstruction representations can effectively transfer in cross-domain anomaly detection scenarios. Specifically, the authors mention that this can resolve multi-scale inconsistency and learn shared anomaly patterns. This raises questions for me, as it is unclear what exactly constitutes shared anomaly patterns. If it merely summarizes information across multiple scales, this approach is already common in many existing multi-scale time series anomaly detection works.

The related work section is divided into reconstruction and temporal anomaly detection parts, which is not a good organizational structure. Reconstruction itself is a common approach in time series anomaly detection. Additionally, the survey of related works on multi-scale anomaly detection and cross-domain anomaly detection is missing.

This work requires pre-training anomaly detection models for D domains in advance, which appears inflexible. There is a lack of reasonable investigation into the transferability across multiple domains.

Furthermore, when the authors mention shared anomaly patterns, is there a requirement for anomaly labels when training domain-specific anomaly detection models?

Overall, the preparation of this work seems rushed, and there are many errors present. For example:
- Lines 317 and 359 reference Table 4.2, which does not exist, indicating a citation error.
- There is a citation error in line 287.

In summary, it is unclear what types of anomaly patterns their method can effectively address. Additionally, there is no visual analysis provided to support this point.

**Questions:**

Please see the weakness part.

---

### Official Review · Reviewer_kR2j · 2025-11-01

**Soundness:** 2
**Presentation:** 3
**Contribution:** 1
**Rating:** 4
**Confidence:** 5

**Summary:**

This article proposes a TSAD model called UniAnomaly, which is a cross-domain adaptive multi-scale framework for unified time series anomaly detection. It aims to address two major challenges: insufficient cross-domain generalization and inconsistent temporal granularity, primarily by performing multi-scale modeling and feature fusion on time series data. The writing and presentation of the article are easy to understand.

**Strengths:**

1. The article is written very clearly and is easy to understand, allowing readers to grasp its core motivation easily;

2. The illustrations in the article are quite refined and present the results well;

3. The article has a great entry point, with cross-domain and cross-scale aspects being important, which is a key issue in the task of temporal anomaly detection.

**Weaknesses:**

1. The contribution of the paper is very limited. Multi-scale and gating mechanisms have been widely explored in time series analysis and temporal anomaly detection, and the combination of existing methods may not provide readers with significantly new insights;

2. The evaluation metrics are relatively limited, and the latest metrics from the community, such as VUS-ROC and VUS-PR, are not introduced, which somewhat diminishes the comprehensiveness of the evaluation results;

3. Among the comparative experiments, there are few new methods from 2025, which further leads to incomplete evaluation and makes it difficult to highlight the effectiveness of the proposed method. It is recommended that the authors include more comparison algorithms.

**Questions:**

1. The article needs to clarify the core differences between this work and other historical works on multi-scale modeling. The current explanation is insufficient, and it is necessary to clearly state the main innovations of this work;

2. It is recommended to improve the experiments. Whether it is baseline methods or other metrics, it is necessary to verify whether the model has indeed achieved a sufficient level of advantage, in order to further highlight the advancement of the proposed method.

---

### Official Review · Reviewer_fCE7 · 2025-11-04

**Soundness:** 3
**Presentation:** 3
**Contribution:** 2
**Rating:** 2
**Confidence:** 5

**Summary:**

This paper proposes UniAnomaly, a reconstruction-based, unsupervised framework for cross-domain time series anomaly detection (TSAD) with multi-scale representation learning. Specifically, UniAnomaly pre-trains on a large collection of diverse datasets to capture shared anomaly patterns, and introduces a multi-scale encoder with patch embedding at multiple resolutions, scale-specific dilated convolutions, and adaptive gated fusion to produce unified representations robust to varying temporal granularity. The key assumption is that multi-domain pre-training can learn domain-agnostic temporal features that transfer effectively to unseen datasets, and multi-scale modeling can handle inconsistencies in sampling frequency by adaptively weighting granularities. The evaluation results on multiple real-world benchmarks demonstrate state-of-the-art performance comparing against comprehensive baselines.

**Strengths:**

1. Critical Problem: The challenges of cross-domain transfer and varying temporal granularity is under-explored yet critical problem.

2. Extensive Evaluation Settings: Compares against 20+ baselines (including DADA, GPT4TS, ModernTCN) using zero-shot + fine-tune protocols and affiliation F1, with systematic ablations studies.

3. Clear Presentation: The paper is well-written with intuitive diagrams to illustrate challenges and proposed framework, making it easy to follow.

**Weaknesses:**

1. Problematic Datasets: The evaluation relies on SMD, MSL, SMAP, SWaT, PSM datasets, which has been shown by the prior art [1] that contain large amount of trivial anomalies, mislabeled events, and global shifts. This makes the SOTA claims meaningless.

2. Domain Leakage for zero-shot setting: Pre-training on ASD, Exathlon (servers) datasets and testing on SMD, PSM (servers) datasets allows model to exploit domain overlap rather than true cross-domain transfer. This may lead to inflated zero-shot performance due to shared data characteristics (e.g., similar sensor types, sampling patterns, and anomaly semantics), which undermines the claim of generalizability to unseen domains.

3. Limited Novelty: Though the combination of multi-resolution patch embedding + per-scale encoders + adaptive fusion is somewhat architectural-wise new, the core concept of multi-scale temporal modeling has been extensively explored in recent time series literature, particularly in foundation models. The use of multi-resolution patch embedding follows from SimMTM [2], per-scale dilated convolutions seems following the TimesNet [3], and adaptive scale fusion via gating is a simplified version of DADA. Without head-to-head ablation comparing with DADA or SimMTM, it is hard to justify the technical contribution.



[1] Wu, Renjie, and Eamonn J. Keogh. "Current time series anomaly detection benchmarks are flawed and are creating the illusion of progress." IEEE transactions on knowledge and data engineering 35.3 (2021): 2421-2429.

[2] Dong, Jiaxiang, et al. "Simmtm: A simple pre-training framework for masked time-series modeling." Advances in Neural Information Processing Systems 36 (2023): 29996-30025.

[3] Wu, Haixu, et al. "Timesnet: Temporal 2d-variation modeling for general time series analysis." arXiv preprint arXiv:2210.02186 (2022).

**Questions:**

See the weakness above and address them.

---

### Meta-Review · Area_Chair_DWCv · 2025-12-14

**Summary:**

The work tackles the problem of cross-domain time-series anomaly detection and introduces a method called UniAnomaly that leverages large-scale pre-training and a multi-scale encoder to learn cross-domain cross-scale anomaly representations. The method is validated on a number of real-world TSAD datasets.

**Reviewer Concerns:**

There are a number of major concerns on the motivation, novelty, experiment design, and justification of the work. No author rebuttal is submitted.

**Reviewer Scores:**

The work receives three reject and one weak reject due to the above concerns. The authors do not respond to the reviewers' concerns.

---

### Decision · Program_Chairs · 2026-01-26

Reject